# A New Model of Mathematics Education: Flat Curriculum with Self-Contained Micro Topics

**Miklós Hoffmann** [1,2,†] and **Attila Egri-Nagy** [3,*,†]

1 Department of Mathematics, Eszterházy Károly University, 3300 Eger, Hungary; hoffmann.miklos@uni-eszterhazy.hu
2 Faculty of Informatics, University of Debrecen, 4032 Debrecen, Hungary
3 Department of Mathematics and Natural Sciences, Akita International University, Yuwa, Akita 010-1292, Japan
* Correspondence: egri-nagy@aiu.ac.jp
† These authors contributed equally to this work.

**Abstract:** The traditional way of presenting mathematical knowledge is logical deduction, which implies a monolithic structure with topics in a strict hierarchical relationship. Despite many recent developments and methodical inventions in mathematics education, many curricula are still close in spirit to this hierarchical structure. However, this organisation of mathematical ideas may not be the most conducive way for learning mathematics. In this paper, we suggest that flattening curricula by developing self-contained micro topics and by providing multiple entry points to knowledge by making the dependency graph of notions and subfields as sparse as possible could improve the effectiveness of teaching mathematics. We argue that a less strictly hierarchical schedule in mathematics education can decrease mathematics anxiety and can prevent students from 'losing the thread' somewhere in the process. This proposal implies a radical re-evaluation of standard teaching methods. As such, it parallels philosophical deconstruction. We provide two examples of how the micro topics can be implemented and consider some possible criticisms of the method. A full-scale and instantaneous change in curricula is neither feasible nor desirable. Here, we aim to change the prevalent attitude of educators by starting a conversation about the flat curriculum alternative.

**Keywords:** mathematics; education; philosophy of education; flat curriculum; micro topics; deconstruction; dependency graph





## 1. Introduction

Mathematics as a scientific discipline has been deductively described and presented in the scientific community for centuries. Even if the presentation is not fully axiomatic, one must possess a solid basis of the preliminary hierarchical system of notions and statements to understand the actual mathematical topic. The theoretical aspects of these educational hierarchical systems, also called learning hierarchies, have been widely studied from the viewpoint of many disciplines for decades (cf. [1–4] and the references therein). However, less attention is paid to the fundamental question of the extent to which it is necessary to maintain this hierarchy in mathematics education from practical and theoretical points of view. Although the deductive approach in mathematics education is becoming increasingly backward, giving way to many modern methodological approaches, such as inquiry-based learning, research-based learning, etc. We still do not seem to be able to go beyond the approach arising from the abovementioned hierarchical structure of concepts and statements in education. This insistence, in turn, makes the mathematical knowledge provided over the school years similar to a monolithic, large structure with a highly interconnected system of concepts, where there is little chance to understand a topic without possessing knowledge from other, previously discussed topics in the curriculum.

There is practically a single entry point into mathematics education. Many children and pupils in school may be frightened when realising the weight, rigour and robustness of this huge building-like structure. Mathematics anxiety, which involves fear, nervousness, or even bodily symptoms related to mathematical activities, is an existing issue in education.

Classical experimental tools exist to measure math anxiety, e.g., Fennema–Sherman Mathematics Attitudes Scales. The outcome of these tests shows that approximately 93% of adult US students indicate that they experience some level of math anxiety [5], 59% of the 15–16-year-old students reported that they often worry that math classes will be difficult for them, 33% reported that they become very tense when they have to complete math homework and a further 31% stated that they become very nervous solving math problems. Math anxiety correlates negatively with PISA math task achievement [6], and there is no significant decrease in math anxiety throughout the years. It generally increases with age among young people.

We aim to provide a significantly and structurally different mathematics education approach: flattening the curriculum and reconsidering the dependencies of topics and notions by introducing a set of self-contained micro topics instead of a single structure. Consequently, our vision of a new system allows for multiple entry points and can contribute to a decrease in math anxiety. As every mathematical and methodological approach has its own philosophical background, our proposal for change also has its foundation inherently embedded in Derrida's deconstruction, which we discuss in detail in an upcoming section. Ultimately, we can characterise our vision by describing mathematical knowledge as a complex decentralised system, but we do not investigate this perspective here.

We claim that the underlying philosophy of mathematics influences how the subject is taught. Curiously, the philosophical assumptions are less critical for actually doing mathematics. In any case, it is fitting to state our assumptions upfront. As a simple classification, we identify two ways of thinking about Mathematics, somewhat similar to [7].

- Mathematics is a beautiful, grand structure of ideas highly connected by powerful abstractions, organised into a hierarchical structure along with the axiomatic-deductive method.
- Mathematics is a set of cognitive tools for efficient thinking. One can conceptualise mathematical thinking as a way to delegate our cognitive load in problem solving to mathematical symbolism.

Roughly speaking, these can be identified as the perspectives of mathematicians and of scientists and engineers. These are often contrasted, especially in the context of pure versus applied mathematics, but we do not see them as contradicting approaches. An engineer can appreciate the beauty of mathematical coherence while simultaneously using a specific method from a narrow branch of mathematics. Similarly, a mathematician can appreciate real-world applications. Whatever professions are associated with these approaches, mathematics education struggles with applying both of them. While trying to demonstrate the importance of problem solving, which is related to the second approach, it cannot break away from the hierarchical structure of the first approach. The result, however, is that the daily struggle to understand pieces of the grand structure draws its breath away from the joy of problem solving. Mathematics education, despite many efforts, is still mostly about building and understanding the hierarchy and connectivity of theoretical notions and statements.

Regarding the ontological status of mathematics, we endorse a noncommittal version of realism. Mathematical objects exist in the sense that they demonstrate some resistance to our inquiries (they 'kick back'), and they have causal power (through the actions of humans). Whether they really exist in some platonic realm or they are merely constructed socially is irrelevant for our discussion. What is crucial for us is that people have access to this reality and that they naturally enjoy thinking about mathematical ideas. Moreover, it is beneficial to society if more people have access to mathematical experiences. The actual mode of existence of mathematical structures can be separated from how we think about

them, as a set of tools or as a grand structure, as outlined above. However, the platonist view tends to conflate these two: objective realism pairs with the axiomatic method. Therefore, what we say will likely confront the platonist view. That is not our goal.

Our claim is that how we teach mathematics and how we present mathematics as a whole depends on which ontological viewpoint we are closer to. Curricula at all levels of education seem to be based mainly on the grand structure view. We argue that this is out of balance. By thinking about a second perspective, we could make mathematical ideas accessible for a wider audience (not just gifted pupils and students with math majors).

## 2. Metaphors for the Process of Learning Mathematics

A metaphor can also describe the gist of our argument. It rephrases the view of the foundationalist description of the nature of mathematics as a solid building. This idea can be traced all the way back to Aristotle. He considered science in his metaphysics as a series of deductions from principles, where principles play a similar role as the foundation of a house. Without putting the necessity and validity of this foundation into question, we argue that the mathematical building of Aristotle is way too high today for a student, and instead of a single giant skyscraper, perhaps we could try to build several smaller buildings in education. However, let us also describe this using an alternative, geographic metaphor.

Climbing a high mountain peak is a dangerous activity. Many become stuck in the middle, and in extreme cases, they may even fall down. The air is thin. However, those few who can reach the summit have a complete view of the countryside, similar to an all-encompassing map—a privileged view.

Walking the country from village to village visiting one hut at a time is a pleasant activity. Spending some time in one place, talking to the people there, then moving to another place is also enjoyable and meaningful. Even if the places wandered do not come together to form a complete map of the countryside, the journey is a valuable life experience. In addition, one can even start to see the big picture as a result of regular hiking.

As these metaphors imply, we suggest that experiencing a collection of small-sized, relatively self-contained mathematical topics can increase the efficiency of mathematics education. We propose this method since we think that the traditional hierarchical structure of the curriculum not only fails to convey the grand structure of mathematical knowledge but also harms most students in the process. Following from the abovementioned anxiety-related problems, our main target readership consists of those teachers who feel that some or most of their students are of mediocre level or labeled as underachievers. Even students who cannot successfully conduct their mathematical studies, lost interest, or suffer from math anxiety could benefit from the proposed method. It is well evidenced in the literature that one of the main hindrances to teacher creativity and successful teaching practice is the curriculum itself: the curricular restrictions [8]. We aim to soften these restrictions in a revolutionary manner drastically. It is not the primary purpose, but we even suspect that a flattened curriculum can lead to a coherent understanding of Mathematics with higher probability. We rely on the natural pattern-matching abilities of the human brain and mind. We claim that this method is even better than giving a ready-made structure, as that does not entail personal knowledge [9]. It is a fundamental change of view, but it may be worth trying after so many failures in mathematics education.

## 3. Philosophical Background—Deconstruction

Mathematical knowledge is organised as a hierarchical system by the deductive axiomatic method. We take this as a fact, though it is conceivable that there might be other principles of organisation. Concepts of the mainstream of 20th century philosophy (of science), such as logicism, positivism, formalism, structuralism and even social constructivism, consider to be mathematics a large, massive building with solid axiomatic foundations. This view has deeply influenced didactical concepts in several countries, and for several decades, the teaching of mathematics has been, and still is, somewhat similar to the construction of this building through numerous school years. However, we challenge the

view that mathematics is best taught according to its inherent structure. In this section, we discuss how the concept of Derrida's deconstruction can change this view and what aspects of thoughts and didactical works can be philosophically supported by deconstruction, yielding a less frightening and less oppressive methical approach for students. The mutually beneficial relationship of mathematics education and philosophy has a long tradition in European history, from the pivotal figures of the ancient Greek classical period—where, in fact, several innovators represented both fields—to the eminent scientists of the 20th century. From Plato to Husserl and from Kant to Bertrand Russell, many philosophers have been influenced by their contemporary (and sometimes by their own) mathematical thoughts and initiatives, while mathematicians have always reflected on the philosophical movements of their period. For a good overview of this cross-fertilisation, see, e.g., [10]. Since the first half of the 20th century, well-known and popular(ised) concepts of the philosophical mainstream have been, among others, logicism (Frege, Russell), logical positivism (Vienna Circle) and formalism (Hilbert). Recently, structuralism has emerged as a view of mathematics (Benacerraf, Saphiro), see, e.g., [11]. What is common in these—sometimes rival—concepts is that they consider mathematics (or science in general) as a large, massive structure, and the aim of our scientific and methodical community is nothing else than to prepare foundations of this structure that are as solid as possible and to build this structure as high as we can. Other recent approaches, such as social constructivism (see [12]) may add critical aspects to the absolutism of mathematics, emphasising its human construction, but the fundamental idea is still to construct something. This view has been and still is the ultimate approach in our education, aiming to construct a large structure called mathematics from preschool (sets and logic) to university. However, as it is well-known since Gödel, the foundation of this building cannot be solid anymore. An alternative way of considering and teaching mathematics can be based on the philosophical concept emerged in the last decades, called deconstruction (or deconstructivism), introduced by Derrida [13] and inspired by Saussure and Heidegger. Deconstruction, in its primal form, is an attempt to criticise and put into question the fundamental concepts of methical approaches and forms of description. One can understand the approach of Derrida from his critical note on classical philosophical (as well as mathematical) view: "we are not dealing with the peaceful coexistence of a vis-a-vis [of notions], but rather with a violent hierarchy. One of the two terms governs the other (axiologically, logically, etc.), or has the upper hand" [13]. In terms of mathematics, the revolutionary approach is to deconstruct this "violent hierarchy" of definitions and theorems and to let teachers and students deal with coexistent notions and concepts without being pressured by the weight of the large structure. We suggest that the widely accepted methods, which organise textbooks and curricula in a quasi-linear or circularly ascending (helix-like) fashion with strict dependencies, are fragile. It is prone to losing students. One may miss a step for some reason, rendering subsequent topics incomprehensible, leading to math anxiety. Instead of these structures, we propose a 'flat curriculum' approach: a collection of self-contained, small size exploratory/constructive problems in no particular order and with as few dependencies as possible. This means the deconstruction of the building, cutting the helix of topics into pieces and preserving the parts (and introducing new ones) that can be studied, understood and enjoyed without constant references to other pieces.

This is, in some sense, indeed contradictory to what we usually think about mathematics. However, let us clarify here again: we are not questioning the structure of mathematics as a scientific discipline. What is beautiful for many mathematicians is that everything is connected to everything and organised into a vast hierarchy. Instead of doubting the importance of this structure, we want to reconsider the way we deal with mathematics at school. We argue that the teaching approach can follow a drastically different way, where we want to separate the study of each sub-area as much as possible from the other areas. In a classical textbook series, the entire mathematical system is built on a few fundamental notions (e.g., function, set), highly unified and synthesised. However, failure and anxiety are almost surely coded into this system: if you do not understand the first notions, you will

understand nothing further due to the continuous reference to these notions. Unfortunately, it is particularly difficult to get the first notions right, since they are abstract (by definition). Studying a set of ideas without constant referencing of their dependencies and without the requirement of forming a basis of developing further ideas may lead to a more relaxed pathway to the same skill set as planned to be achieved by regular curricula. Contrary to the standard educational approach, we are not trying to find "the Truth" but to study coexistent "truths".

Considering this view, some further connections to and directions of philosophy (of science) may come to our mind. Questioning the need for absolute foundations and the rationale behind multiple, co-existing alternatives may lead us to fields such as anti-foundationalism (for a good overview, see, e.g., [7,14]) and pluralism of mathematics [15]. Perhaps meaning is not derived from a sound base but the interactions inside a network of ideas? Could it be that mathematics is also non-foundational? Do we just have a persistent illusion since Euclid? Here, we focus primarily on education; thus, the study of these questions and connections is beyond the scope of this paper.

## 4. Implementation

Once we grounded our new approach in theory, the fundamental question naturally arises. How to put it all into practice? How to implement a deconstructed, flattened curriculum?

First, we need to produce a set of micro topics from the existing curriculum. These should be independent monadic structures. Second, we need to bring in new topics, possibly those thought to be inaccessible in school mathematics. The alleged lack of reachability is due to the simple fact that recent developments, by definition, are at the end of the dependency chains of concepts. Once we give up the hierarchical order, we have no reason for excluding newer mathematics.

Let us consider the dependency graph of notions and concepts in the mathematics curriculum to understand this approach better. It is similar to the graph in [16], where vertices denote notions and concepts, and an edge connects two vertices if they are dependent, meaning that the understanding of one notion/concept is required to understand the other one. This is a directed graph, and hopefully, it is acyclic, although this latter property is not entirely assured and evidenced in every textbook. Rephrased in this graph-theoretical language, we aim to make this graph relatively sparse and to discuss as many notions/concepts (i.e. vertices) as possible. We intend to identify and separate a more extensive set of so-called micro topics, small subgraphs with as few incoming edges as possible but with several potential outgoing edges.

### 4.1. Creating Micro Topics

A *micro topic* is a piece of mathematical knowledge that can be presented and understood in one session. The length of a session depends on the age group, so we cannot specify it precisely. Moreover, a micro topic is self-contained. We claim that mathematics can be presented in this monadic format, even though the subject is inherently hierarchical. Of course, a significant amount of work is needed to create this presentation format. By using the following algorithm, we can pick any mathematical idea, and then, we need to take the following steps:

1.  Listing dependencies;
2.  Substituting dependencies, i.e., removing incoming edges; and
3.  Installing hooks, i.e., indicators of outgoing edges.

First, we *list the dependencies* of the concept and construct a dependency graph. We need to clarify what objects we need to mention when expressing the chosen idea. In other words, we need to identify the incoming arrows in the graph. This is different from the linear chain of dependencies of a traditional curriculum since we put the chosen topic in the center and explore the connecting ideas. It is a standard feature of mathematical

monographs to show a dependency graph and to suggest several paths to traverse it. Our approach simply takes this custom one step further.

The next step is *substituting dependencies*. Although possible in some cases, we cannot simply erase all dependencies. That would result in unreasonable requests to believe something without explanation, and understanding through explanations is what we are trying to facilitate. However, human thinking has an enormous capacity for compressing ideas with analogies, images, stories, metaphors and intuitive descriptions. A classic example is defining continuity by saying that we can draw a curve without lifting the pen. Once deep we are deep into calculus, we need the rigorous $\varepsilon$-$\delta$ definition, but the intuitive definition serves as a great entry point. Another example would be a statistics course, where calculus is not a prerequisite. Suppose we want to go from discrete probability distributions to continuous ones. Traditionally, this assumes background knowledge in integral calculus. However, the exact definition of the Riemann integral can be substituted with the intuitive explanation of increasingly thinner bars in the histogram. With a bit of squinting, we can see a continuous curve. In general, finding substitutes such as this may require creativity.

Finally, we install hooks, the outgoing dependencies. The monadic nature of a micro topic facilitates the sense of achievement but does not aim to limit the learner to cut off the outgoing connections. Instead, we want to maintain their potential. These can be represented as intriguing questions, without referring to other micro topics explicitly but letting the pupils themselves discover these connections.

### 4.2. Prior Art

It is often remarked that teaching mathematics and conducting research may require a different set of skills. Gian-Carlo Rota remarked that "gifted expositors" of mathematics could be rarer than successful researchers [17]. He also added that "if you wish the reader to follow rather than decipher, the linear deductive technique of exposition is the worst", but beyond pointing to the engineers, the alternatives were not explored.

Developing these types of micro topics is also not a new idea. It has been done several times in popular science math books. The classical example is the works of Martin Gardner [18], but there are numerous other examples (for instance [19,20]). Sadly, these works seldom make it into the classroom. Moreover, regular textbooks never take this approach. Why is this the case? Is there a true educational, cognitive science-based reason, or is it simply just a continuing misunderstanding that real work cannot be entertaining? We believe in the latter, but this a future task for research. Here, we aim to change the general attitude, so teachers will be more open to experiment with this approach.

In his seminal work entitled Proofs and Refutations, Imre Lakatos also developed some problems of mathematics in the form of dialogues—these pieces can also be considered as micro topics in our context [21]. In the form of small maths gems, micro topics often also appear at public promotional presentations in science centers; math museums; and other outreach events, such as Researchers' Night. A crucial part of these events is to provide the audience with understandable, exciting, one-of-a-kind lectures on some mathematical topics without requiring specific preliminary knowledge or special training.

### 4.3. Examples of Micro Topics

Here, we provide some examples of micro topics. These already proved successful from the practice of the authors. We emphasise that our goal here is not to process the whole of mathematics and develop yet another entirely new curriculum but just to shed light through a few examples of what we mean by a specific micro topic.

#### 4.3.1. Group Theory in Primary School

As an example for bringing advanced mathematics into the classroom, we mention a successful educational endeavor. Can we explain the basic concepts of group theory to primary school pupils? We think the answer is yes, assuming enough thoughtful

preparation. Within the program 'Mathematicians in Schools' by CSIRO, Australia's science and research agency, we had a chance to trial this idea in an extracurricular session for 15 students of mixed ages from 9 to 12. Naturally, the class format has to be different from a university lecture for such an age range. The session has to include some hands-on experiences.

The main idea of the session comes from an informal definition of symmetry for a popular science mathematics book.

> "You could think of the total symmetry of an object as all the moves that the mathematician could make to trick you into thinking that he hadn't touched it at all".

[22]

We used large-sized cut-out geometric shapes laid out on the floor to demonstrate that a symmetry operation's result is not distinguishable from doing nothing. Half of the class was asked to turn away, while the other half observed the instructor's action. Then, the non-observing half had to decide whether the instructor touched the shape or not. The recognition of symmetry operations led to counting their numbers for a given shape. Again, we relied on a crisp formulation of the idea of a symmetry group.

> "Numbers measure size, groups measure symmetry".

[23]

The children were fascinated by the idea of the identity operation. They found it amusing that doing nothing is a necessary transformation. The counting of symmetries was facilitated by labeling the corners of the shapes. Finally, the symmetries were put into a composition table to track how they combine. We intentionally avoided calling it a multiplication table since that would have caused some confusion. The session ended with the students recognising patterns in the composition table of the cyclic group.

The books' quotes show that much work is needed to make a short session possible. For both authors, the concise informal definitions are the distilled summaries of years of work. Creating micro topics will require a large amount of work, but some of them might have been completed already, as the example shows.

Group theory also allows for thinking about the limits of the micro topics method. Can we summarise quickly the ideas involved in the Monster Group? A short book can [24], which is still far from being a micro topic.

### 4.3.2. Differential Geometry of Curves and Surfaces in Secondary School

Another example of bringing advanced micro topics into the classroom is introducing some basic concepts of differential geometry of curves and surfaces. This micro topic has also been proven successful in several extracurricular classroom events in various secondary schools.

We need to avoid the application of exact calculus. Nevertheless, keeping our discussion almost exclusively on geometry, we can quickly introduce various fundamental notions. The first notion must be the naive concept of differentiable parametric curve, where differentiable simply means something that can be drawn smoothly, with no cusps. The parameter can be the time as we travel along the curved path. Connecting two points of this curve by a straight line and moving one point towards the other along the curve, we obtain the notion of the tangent line, where students can discover the fundamental difference between the elementary definition of the tangent of a circle and the general notion. This finishes the substitution for the missing calculus.

The central notion is the curvature. Suppose we draw a couple of convex curves touching the same line at a point with different (higher and higher) curvatures. In that case, students can easily grasp the "feeling" of being less or more curved: it is soon discovered that the more curved curves differ more radically from the original direction than the less curved ones. Here, we can also refer to driving a bicycle on a curved road. Using the concept of a parameterised path, the curvature can be jointly discovered and introduced

as the deviation of tangent lines at two neighboring points at the same distance along the different curves, measured by the angle of the two tangent line.

The bicycle path analogy can also lead to the discovery of curves with constant curvature (no need to move the handlebar at constant speed), inflection points (a moment in an S-shape path where the handlebar is in its original position), etc.

After understanding the notion of curvature of curves, the Gaussian curvature of surfaces can also be introduced. Students can imagine a hilly landscape, where at one point, there are infinitely many directions paths, and in each direction, one can measure the curvature of the actual path at that point (normal curvatures). It is evident that—if the landscape is flat and not a perfect sphere—one can find the steepest path and, on the contrary, the least sloping path and their curvatures (principal curvatures). Although it is generally a big surprise for students that a Gaussian curvature is the product (and not the average) of these principal curvatures, the further discovery of the meaning and geometric consequences of this product (sign, being equal to zero) soon convinces them of the benefits of multiplying the two extrema.

### 4.4. Creative Substitution

Substitutions can be made just for a single purpose, or they can be more systematical. A successful example of popularising mathematics is [25], where the underlying metaphor, mathematics-as-cooking, spans the whole book. This is a notable example, since it manages to convey the basic ideas of category theory, which is often considered too abstract even by mathematicians. Clearly, this is a result of years of thinking or rather a way of living life. Powerful substitutions are not easy to find, but they are possible.

### 5. Flattening the Curriculum

From the viewpoint of the foundations of mathematics, nothing is possible without set theory (or any other foundational part of mathematics). Furthermore, by definition, no step in the reasoning process can be left out in logical deduction. We argue that the presentation of a topic in education must not follow this logical order and the order of construction. Set theory, being foundational, is the starting point of the standard curriculum. However, should it be the first to present? Deductive logic says so, but the history of mathematics shows that mathematical ideas could be discussed before set theory was worked out. One might argue that it gives a unifying abstract language for the whole subject, but we also know that abstraction does require mental effort. The Wason selection task demonstrates that people reason better in familiar social contexts than with abstract objects [26,27]. The question is, what is the right balance between the logical ideal and cognitive accessibility?

The fundamental problem is that we still insist that the strong interrelationship of mathematical fields and notions must be present in mathematics education. Mathematics is provided to students as a single massive logical structure, an unquestionable monolithic building. Even if the pedagogical and methodological experiments of the last decades, especially inquiry-based learning, try to soften this deductive approach, we cannot break away from this epistemic viewpoint. Inquiry-based learning and similar methodological initiatives try to show something new within a specific subfield to make mathematics more "user-friendly", but they do not question the strict interdependence of the subfields, the global logical structure of mathematics and the necessity of presentation of this structure.

However, human thinking has several other modes of operation, e.g., metaphorical thinking. There are different ways of thinking about the same idea [28]. While logical formalism can precisely fix the meaning, it may make more nuances and the more tenuous connections become less apparent. A monadic/flat curriculum can maintain these nuances, as the different meanings can play a role in different topics.

We argue that flattening the curriculum can be an alternative educational approach to mathematics. To flatten the curriculum, the key idea is to make the dependency graph of the presented subfields as sparse as possible. The ultimate goal of flattening the curriculum

is to provide more than one entry point to make it more accessible for students who have missed or understood less of the previous topics.

Programming education has a similar setup. The traditional approach starts at the bottom of the computing stack, e.g., the binary representation of numbers, digital logic, and computer architecture, and then proceeds to the level of the operating system and eventually to application programming in a higher-level language. This is not a possible approach in coding for kids. Additionally, on the undergraduate level, learning higher-level programming languages for students not majoring in computer science may be obstructed by the assumed background knowledge of the inner workings of computers. With a bit of thinking and careful language choice, this can be avoided.

## 6. Possible Criticisms

Here, we are trying to answer some of the possible criticisms preemptively. We expect resistance to the idea of a flat, multi-entry curriculum. Even if these criticisms become straw man arguments, they might help understand the concept deeper.

### 6.1. Assessment

Grading and assessment are always an issue. Our proposal implies that each school can have its own randomised order of topics (even when assuming a centrally curated collection of topics). How can we then compare the performance characteristics of schools? How do we evaluate the performance of individual students in the bite-size topics? Indeed, switching to a flat curriculum while keeping the system of standardised tests would be a disastrous combination. Developing assessment methods along the flat curriculum could require even more work than creating micro topics. However, if we are not happy with the learning efficiency of standardised tests, this work is unavoidable anyway.

### 6.2. Lack of the Complete Picture of the Subject

One might argue that having a coherent worldview is vital to dealing with information overload in today's world. Such a perspective should be the primary task of education, but the proposed flat curriculum method fails to do that. Our rejoinder is that giving the ready-made hierarchical view of mathematics also fails to provide that coherent picture reliably. Moreover, a complete, hierarchically structured knowledge may not be the only way to provide a helpful perspective. It can be in the process of approaching problems, in the strategy, in the way of relating to challenges.

Are the micro topics just another pure discovery learning method [29]? No, we do not measure the success of a flat curriculum by the students' ability to reconstruct the whole picture of mathematics. However, we assess success by the quality of the relationship with mathematical thinking that they develop. Roughly speaking, the monolithic curriculum frequently yields students feeling left out and developing math anxiety, while the grand picture is not guaranteed even for students who can keep up. We think that a flat curriculum with multiple entry points can lead to a better experience of mathematics. We do not claim that micro topics automatically lead to a complete understanding. However, we suspect that the human brain's pattern-matching abilities will naturally spring into action once the anxiety is released. Additionally, a flat curriculum has plenty of opportunities for guided discovery. Of course, the proof of these claims requires future empirical investigations.

## 7. Conclusions

We described an alternative method for organising a mathematics curriculum. Instead of the strictly hierarchical and quasi-linear or helix-like traditional school curriculum, we promote micro topics, monadic pieces of mathematical knowledge to improve accessibility to mathematics. From a philosophical point of view, this approach is based on the principle of deconstruction.

Micro topics are proven successful in out-of-classroom events, such as the outreach event of the Researchers' Night. There, a topic was presented without the requirement

of specific preliminary knowledge. Why not apply this approach in the curriculum? We argue that we made a mistake by not using this micro topic approach instead of the "serious" (i.e., a hierarchically established large system of) mathematics in the classroom. A set of largely independent micro topics provides a chance for multiple entry points to mathematics while not eliminating the possibility for students to discover connections. The directed dependency graph of notions and concepts of these micro topics is much sparser in incoming edges than in a usual curriculum. However, outgoing directions, called hooks, preserve the potential of discovery of connections. This way, the curriculum is flattened. Instead of a large monolithic structure, it contains various, horizontally spreading smaller subfields, decreasing the feeling of the monumentality of a hierarchical structure and hopefully decreasing math anxiety.

We are fully aware of the radicality of this proposal, and it is clear that much work is needed to create a flattened curriculum and to investigate its benefits and possible disadvantages. Especially, for this reason, it is critical now that more researchers and educators open their minds up to this way of thinking. This is what we try to achieve with this paper.

**Author Contributions:** The fundamental ideas presented in this paper were developed independently by the two authors. At a chance meeting, M.H. described the climbing metaphor and suggested writing it up in a more detailed way. A.E.-N. and M.H. contributed equally to writing the paper. All authors have read and agreed to the published version of the manuscript.

**Funding:** This research was partially funded by the internal research grant of Akita International University and by the MTA KOZOKT2021-44 Public Education Development Research Program of the Hungarian Academy of Sciences.

**Conflicts of Interest:** The authors declare no conflicts of interest. The funders had no role in the writing of the manuscript.

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
