# Peer review of "A New Model of Mathematics Education: Flat Curriculum with Self-Contained Micro Topics"

_philosophies, doi:10.3390/philosophies6030076_

Round 1

Reviewer 1 Report

This is a very interesting paper, which is a must read for anyone who cares about mathematics education. Above all, the suggestion as to “flattening curricula by developing self-contained micro topics” sounds intriguing. There is no doubt that this suggestion is well thought out and clearly argued for in compact manner.

Insofar as this paper contributes both negatively and constructively to mathematics education, it may still be necessary to think further about the balance between the two. If the authors seriously want “climbing a high peak of a mountain” to be superseded by “walking the country”, they should consider the possibility that the readers might be happier to know more about the possible consequences of the proposed reform.

At this juncture, it seems ironical that the author might be championing a theory of double truth. Tersely put: Do authors propose their new approach for all humans including the future Field medalists, or just for mediocre students?

Depending on the authors’ reply to my question, I think, we might examine further the different components of their proposal. For example, what is it for presenting their approach as radical as possible by introducing Derrida? Wouldn’t it be enough to discuss some of the leading positions in recent philosophy of mathematics? Simple curiosity to another issue: In view of the huge market for popular mathematics books, how are we to explain their failure to change the mathematics curriculum?

I am going to sit down and study very carefully section 4 in the near future. Because I tend to be sympathetic with the authors’ following remark: “To flatten the curriculum, the key idea is to make the dependency graph of the presented subfields as sparse as possible. The ultimate goal of flattening the curriculum is to provide more than one entry point to make it more accessible for those students who have missed or understood less of the previous topics.”

There are some minor problems in section 3 of the paper that could irritate professional (in the worst sense of the word) philosophers. For example, Russell was not a logical positivist.

We have reason to think much more deeply about the authors’ climbing metaphor. But I would like to advise them, if I may, not to sell too much of it.

Author Response

We were heartened by the positive comments, since we are bracing for harsh criticisms (what we met in several informal conversations with mathematicians).

We clarified the target audience, which could help to see the positive results the approach may bring.

Regarding mentioning Derrida. Our proposal does require radical re-evaluation of ideas, thus we find the reference appropriate. We are aware of the possibility that this strategy will backfire, but there is also danger in downplaying the radicality of the approach. In any case, we toned down the connection a bit.

Regarding the disconnect between the popular science works and the standard curriculum, we have no clear explanation. Our best guess is that the reason is simply a misunderstanding. Our personal teaching experiences show that whenever we introduce topics from those books, we have success in the classroom. We added some reflections on this.

"I am going to sit down and study very carefully section 4 in the near future." - we have never seen a sentence like this in a review. This is the best reward that authors can get. Thank you!

We corrected the mentioned mistake.

Reviewer 2 Report

The manuscript explores a new model of mathematics education. 

  • Abstract: reviewer suggests re-writing this part of the manuscript within 200 words limitation
    • Reviewer suggests to improve in a better way the background part of the abstract
  • Manuscript needs a moderate english and text-style chage

Author Response

We extended the abstract to match the 200 words ideal size. Now the abstract better describes the content of the paper.

We went through the text and improve the style.